

# Geographical distribution of two major quarantine fruit flies (*Bactrocera minax* Enderlein and *Bactrocera dorsalis* Hendel) in Sichuan Basin based on four SDMs

Yanli Xia[1,*], Jinpeng Zhao[2], Jian Ding[3], Ke Xu[4], Xianjian Zhou[5], Mian Xiang[5], Huiling Xue[1], Huan Wang[1], Rulin Wang[2] and Yuxia Yang[5,*]

[1] School of Food and Bioengineering, Chengdu University, Chengdu, China
[2] Sichuan Provincial Rural Economic Information Center, Chengdu, China
[3] Sichuan Science and Technology Exchange Center, Chengdu, China
[4] Sichuan Horticultural Crop Technology Extension Station, Chengdu, China
[5] Sichuan Provincial Key Laboratory of Quality and Innovation Research of Chinese Materia Medica, Sichuan Academy of Traditional Chinese Medicine Sciences, Chengdu, China
[*] These authors contributed equally to this work.

Corresponding author
Yuxia Yang, yangyuxia-7@163.com

## ABSTRACT

Both *Bactrocera minax* and *Bactrocera dorsalis* are phytophagous insects, and their larvae are latent feeders, which cause great damage and economic losses to agriculture production and trade. This study aimed to provide a scientific reference for researching and developing the feasible countermeasures against these two pests. Based on the distribution data of *B. minax* and *B. dorsalis* in China, obtained from the Chinese herbaria, investigation and literature. Four niche models (Garp, Bioclim, Domain, and Maxent) were used to analyze the key environmental factors affecting the distribution of both pests and to build prediction models of the potential distribution in Sichuan Basin. Combined with two statistical standards, area under the receiver operating characteristic curve (AUC) and Kappa, the validity of prediction models were analyzed and compared. The results show that: the average AUC values of the four models are all above 0.90, and the average Kappa values are all above 0.75, indicating that the four models are suitable for predicting the potential distribution area of *B. minax* and *B. dorsalis*. The annual range of temperature, the mean temperature in the driest quarter, the mean temperature in the warmest quarter, the annual precipitation, and the precipitation in driest month are the key environmental factors affecting the distribution of *B. minax*, while the mean diurnal temperature range, the mean temperature in the driest quarter, the seasonal temperature variations and the precipitation in driest month affect the potential distribution of *B. dorsalis*. The suitable areas for *B. minax* are mainly concentrated in the eastern of Sichuan Basin, while the suitable areas for *B. dorsalis* are concentrated in the southeastern. Except for the Bioclim model, the highly-suitable area for both pests predicted by the other three models are all greater than $15.94 \times 10^4$ km$^2$ and the moderately-suitable areas are greater than $13.57 \times 10^4$ km$^2$. In conclusion, the suitable areas for both pests in Sichuan Basin are quite wide. Therefore, the relevant authorities should be given strengthened monitoring of both pests, especially in areas with high incursion rates.

# INTRODUCTION

Fruit flies are insects belonging to the order Diptera and the family Tephritidae, characterized by one pair of functional wings and one pair of reduced halteres, a diverse range of habitats and feeding behaviors, and a significant impact on various ecosystems and human activities (*Elekçioğlu, 2013*). The common species in China are *Bactrocera minax* Enderlein, *Bactrocera dorsalis* Hendel and *Tetradacus tsuneonis* Miyake, which are all international quarantine pests (*Wang & Zhang, 2009*). *B. minax* is a native species (*Liang, Zhang & Xu, 1989*), mainly distributed in southern citrus regions, which can harm a variety of citrus fruits, such as sweet orange, grapefruit, lemon, bergamot (*Zhang, 2007*). The host species of *B. dorsalis* is complex, including more than 250 kinds of trees, vegetables and flowers, such as mango, pomegranate, lime and so on (*Liang et al., 2003*). *B. dorsalis* is a non-native species (*Zhang et al., 2022*), but it is widely distributed. In addition to the southern citrus regions, it has also been found in central China in recent years. Both *B. minax* and *B. dorsalis* are invasive and polyphagous pests of horticultural crops, which have caused great damage and economic losses to agricultural production and agricultural trade (*Ye et al., 2022*).

*B. minax* occurs one generation a year and overwinters in the soil as a pupae. The eclosion time of adult insects varies with year and place, mainly affected by the mild atmosphere and soil moisture, the optimal temperature is about 22 °C and the soil moisture is between 15% and 20%. In Sichuan Basin, the eclosion time of *B. minax* begins in late April, and gradually ends in mid-June, which is generally consistent with the phenological period from flowering to fruiting (*Tang, 2012*). *B. dorsalis* usually occurs three to five generations a year in subtropical areas and overwinters in the soil as an old larvae or pupae (*Liang et al., 2003*). The development stage and the hatching rate of eggs are affected by temperature and humidity, and it is very poor when the temperature is lower than 16 °C or higher than 36 °C, and the relative humidity is lower than 40%. The larval stage is related to temperature and host, ranging form five to ten days at $28 \pm 1$ °C and eight to eleven days at $25 \pm 1$ °C. The pupal stage is greatly affected by soil water content and temperature, the mortality rate is higher when the soil water content is lower than 40% or higher than 80%, and the pupal stage doubles and the emergence rate decreases when the temperature is lower than 18 °C (*Zhou, 2016*).

The ecological characteristics of crop diseases and pests, such as growth, reproduction, overwintering and distribution, are closely related to environmental conditions, especially climate conditions. Therefore, climate change have a great impact on the generation, wintering northern boundaries and the distribution range of crop diseases and pests (*Li et al., 2010*). According to previous studies, the annual area of pests in China is between 200 million and 230 million square kilometers, more than twice as much as arable (*Zhou et al., 2013*). Citrus is the most important fruit in Sichuan Basin, its area and yield are increasing
year by year. By the end of 2020, the citrus planting area reached $3.39 \times 10^5$ hectares and the yield reached $4.89 \times 10^6$ tons (*Wang & Du, 2022*). In view of the importance of citrus to the economy of Sichuan Basin, it is vital to control pests.

Species distribution models (SDMs) use certain algorithms to predict the potential distributional range of species based on actual distribution points and environmental factors. The selection of environmental factors, the migration ability of species and the types of environmental factors have an important impact on the simulation results (*Gobeyn et al., 2019*; *Van Eupen et al., 2021*). At present, The SDMs used to predict the potential distribution of species include Climex, Garp, MaxEnt, Bioclim and Domain (*Li et al., 2013*; *Xu, Peng & Peng, 2015*), many scholars used these models to predict the potential distribution of pests, such as *Bemisia tabaci* (*Li et al., 2023*), *Wasmannia auropunctata* (*Xu et al., 2023*), *Locusts* (*Ju et al., 2022*), *etc.*

Recently, some scholars have studied *B. minax* and *B. dorsalis*, mainly focuses on biology (*Sánchez-Rosario et al., 2022*; *Akami et al., 2022*), ecology (*Diesner et al., 2022*), quarantine and treatment technology (*Maktura, Paranhos & Marques-Souza, 2021*; *Hou et al., 2021*), integrated prevention and control technology (*Shelly et al., 2022*; *Yoshida et al., 2021*), but there is no research on theirs geographical division in China. In this article, we predicted the distribution of *B. minax* and *B. dorsalis* with SDMs, obtained the key environmental factors affecting theirs geographical distribution, and delimited theirs suitable area. This study aimed to provide a scientific reference for researching and developing the feasible countermeasures against *B. minax* and *B. dorsalis*, reduce the serious economic loss caused by the them, and ensure the steady development of the fruit, vegetable and flower industry in China.

## MATERIALS AND METHODS

### Sample distribution data

Occurrence records of *B. minax* and *B. dorsalis* in China were obtained from various sources, such as the Global Biodiversity Information Facility (*GBIF, 2023a*; *GBIF, 2023b*), Centre Agriculture Bioscience International (*CABI, 2023*) and literature (*Zhao et al., 2006*; *Zhou et al., 2010*; *Fan, Yang & Yin, 2013*; *Wu et al., 2014*; *Li et al., 2017*; *Xu et al., 2017*; *Deng, Huang & Dong, 2018*; *Li et al., 2019*; *Zhang, Mao & Du, 2019*; *Cui & Liu, 2020*; *Fang et al., 2020*). The longitude and latitude of occurrence records picked up by Google Earth were converted into decimal after removing the repeated distribution points. In order to reduce the impact of spatial self-correlation, the ENMTOOL software was applied to process occurrence records to ensure that each grid (size: 30 arc-seconds) contains only one distribution point (*Wang et al., 2019*; *Wang et al., 2020b*). After the above steps, two hundred and fifty-one valid records were retained to build the prediction model for *B. minax*, and fifty-seven valid records were retained for *B. dorsalis* (Fig. 1).

### Environmental factors used in models

The growth and development of insects are closely related to the climatic conditions in which they live (*Li et al., 2020*). Therefore, nineteen bioclimatic factors and elevation downloaded from WorldClim (https://www.worldclim.org/) were selected as initial
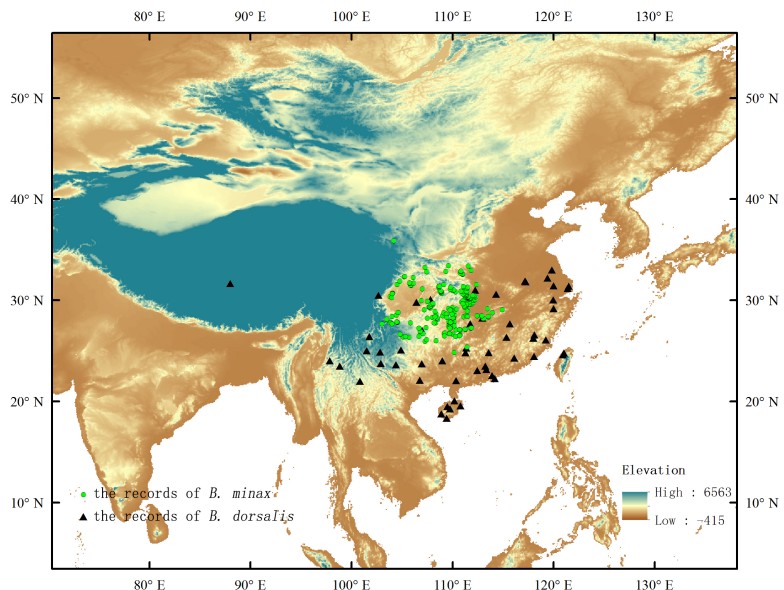

**Figure 1** Existence records of *B. minax* and *B. dorsalis* in China.

environmental factors. In order to eliminate the correlation among the nineteen bioclimatic factors, the following operations were carried out. First, ArcGIS was used to extract the attribute values of nineteen bioclimate factors corresponding to the occurrence records of *B. minax* and *B. dorsalis*. Second, the percent contribution of nineteen bioclimatic factors was tested by the jackknife method, and the factors with zero contribution rate were excluded. Third, correlation analysis was performed for the retained bioclimatic factors (Fig. 2). For the two bioclimatic factors with pearson correlation coefficient greater than 0.85, the factor with lower percent contribution value in the jackknife test was excluded. After the above three steps, thirteen environmental factors were retained to build the prediction model for *B. minax*, and six environmental factors were retained for *B. dorsalis* (Table 1).

## Verification of model accuracy

Receiver operating characteristic (ROC) and Kappa statistics are common methods for evaluating the accuracy of SDMs (*Xu, Peng & Peng, 2015*). ROC curve evaluation method uses the area under curve (AUC) enclosed by ROC curve and horizontal coordinate to evaluate the prediction precisions. The value of AUC ranges is 0 to 1, and the closer the value is to one, the more accurate the prediction result will be (*Liu et al., 2021*). Kappa statistics comprehensively considers the species distribution rate, sensitivity and specificity, and the value ranges is −1 to 1. When the value is greater than 0.75, the consistency is good, and when it is less than 0.4, the consistency is poor (*Wang, 2006*).
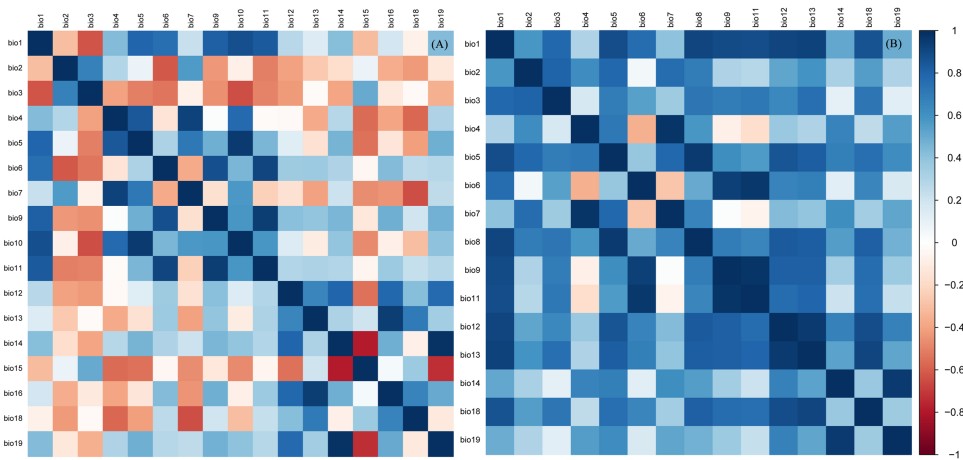

**Figure 2** Correlation analysis of environmental factors affecting potential distribution of *B. minax* (A) and *B. dorsalis* (B).

**Table 1** Environmental factors affecting the potential distribution of *B. minax* or *B. dorsalis*.

| Environment factors | | | Species | |
|---|---|---|---|---|
| Code | Variable | Unit | *B. minax* | *B. dorsalis* |
| Bio2 | Mean Diurnal Range (Mean of monthly (max temp - min temp)) | °C | ✓ | ✓ |
| Bio3 | Isothermality(Bio2/Bio7) (×100) | / | ✓ | ✓ |
| Bio4 | Temperature Seasonality (standard deviation ×100) | / | | ✓ |
| Bio5 | Max Temperature of Warmest Month | °C | ✓ | |
| Bio6 | Min Temperature of Coldest Month | °C | ✓ | |
| Bio7 | Temperature Annual Range (BIO5-BIO6) | °C | ✓ | |
| Bio9 | Mean Temperature of Driest Quarter | °C | ✓ | ✓ |
| Bio10 | Mean Temperatureof Warmest Quarter | °C | ✓ | |
| Bio12 | Annual Precipitation | mm | ✓ | |
| Bio13 | Precipitation of Wettest Month | mm | ✓ | |
| Bio14 | Precipitation of Driest Month | mm | ✓ | ✓ |
| Bio15 | Precipitation Seasonality (Coefficient of Variation) | mm | ✓ | |
| Bio18 | Precipitation of Warmest Quarter | mm | ✓ | |
| Alt | Elevation | m | ✓ | ✓ |

## Prediction model and modeling process
### Maxent model

MaxEnt software running Maxent model was downloaded from official website (https://biodiversityinformatics.amnh.org/open_source/maxent/). In order to improve the accuracy of the Maxent model, we optimized the model by adjusting regularization multiplier (RM) and feature combinations (FCs). The FCs contains five characteristics, such as linear (L), quadratic (Q), hinge (H), product (P) and threshold (T). The operations were as follows: The RM range was set to 0.5 to 4, and the step was 0.5. The FCs was set

as L, LQ, H, LQH, LQHP and LQHPT, respectively. Therefore, 48 different combinations of RM and FCs were established and the combination with the smallest AICc value was selected as the optimal model. Meanwhile, other parameters of the optimal model were set as follows: 25% distribution points were selected as the test set, 75% distribution points were selected as the training set, the maximum number of iterations was set to 500, the maximum number of background points was set to 10,000, and the number of repetitions was set to 10 (*Zhao et al., 2022*; *Guo et al., 2022*).

### *Garp model*
Desktop Garp was downloaded from Lifemapper (https://www.pragma-grid.net/software/lifemapper/). Retaining the default settings except that the number of runs was set to 20 times, that was, the convergence limit and the max iterations was set to 0.01 and 1,000 (*Peterson, Papes & Eaton, 2007*), respectively. By superimposing the result of 10 optimal models in ArcGIS, the geographical distribution prediction map of species can be obtained (*Anderson, Lew & Peterson, 2003*).

### *Bioclim and Domain model*
DIVA-GIS software running Bioclim and Domain model was downloaded from DIVA-GIS official website (https://www.diva-gis.org/). The training data set in shp format was added to DIVA-GIS, then environmental factors were converted into grd format to generate stack data set. Added a stack file in the modeling Bioclim/Domain module and selected the two models for prediction (*Wang et al., 2020a*).

## Suitable area division and area calcultaion
The natural break point method was used to classify the predicted results, and the potential distribution was divided into four types of suitable area: the unsuitable area, the lowly-suitable area, the moderately-suitable area, the highly-suitable area (*Zhang et al., 2023*). In order to avoid over-interpretation the prediction results of the Domain model with natural break point method, the research referenced *Carpenter, Gillison & Winter (1993)* and combined with the matching degree between the output of Domain and the actual distribution of *B. minax* and *B. dorsalis* to adjust the suitable areas predicted by the Domain model. After reclassification, the pixel counts of different suitable areas were counted through "attribute-symbol system-unique value", then the area of each suitable area in Sichuan Basin was calculated base on the number of pixels.

## RESULTS
### Evaluation of prediction accuracy of different models
Ten sets of training data and test data were used to conduct ROC analysis and Kappa statistics for four niche models. As can be seen from Table 2, the average AUC values of four niche models (Maxent, Garp, Bioclim and Domain) are 0.985, 0.916, 0.933 and 0.950, all of which are higher than those of random model (AUC $=0.5$). The Kappa values are 0.825, 0.753, 0.768 and 0.803, all of which are greater than 0.75, indicating that the four models have significant consistency and high prediction accuracy.

**Table 2  Potential distribution of *B. minax* (A–D) and *B. dorsalis* (E–H) simulated by the MaxEnt, Garp, Bioclim and Domain models.**

| Groups | Area under receiver operating characteristic curve (AUC) | | | | Consistency test statistics (Kappa) | | | |
|---|---|---|---|---|---|---|---|---|
| | Maxent | Garp | Bioclim | Domain | Maxent | Garp | Bioclim | Domain |
| 1 | 0.985 | 0.925 | 0.916 | 0.922 | 0.831 | 0.754 | 0.815 | 0.801 |
| 2 | 0.981 | 0.923 | 0.935 | 0.944 | 0.821 | 0.72 | 0.754 | 0.796 |
| 3 | 0.984 | 0.911 | 0.947 | 0.956 | 0.83 | 0.76 | 0.763 | 0.81 |
| 4 | 0.977 | 0.907 | 0.934 | 0.968 | 0.856 | 0.751 | 0.768 | 0.821 |
| 5 | 0.989 | 0.911 | 0.927 | 0.978 | 0.832 | 0.732 | 0.746 | 0.81 |
| 6 | 0.992 | 0.928 | 0.917 | 0.964 | 0.801 | 0.744 | 0.757 | 0.814 |
| 7 | 0.997 | 0.906 | 0.942 | 0.922 | 0.792 | 0.715 | 0.779 | 0.783 |
| 8 | 0.981 | 0.913 | 0.927 | 0.937 | 0.827 | 0.806 | 0.782 | 0.788 |
| 9 | 0.976 | 0.918 | 0.937 | 0.958 | 0.831 | 0.769 | 0.743 | 0.814 |
| 10 | 0.988 | 0.922 | 0.945 | 0.949 | 0.828 | 0.774 | 0.775 | 0.792 |
| Average | 0.985 | 0.916 | 0.933 | 0.950 | 0.825 | 0.753 | 0.768 | 0.803 |

## Potential distribution of *B. minax* and *B. dorsalis* simulated by four models

The prediction results of four ecological niche models on the potential distribution of *B. minax* and *B. dorsalis* in Sichuan Basin are shown in Fig. 3. The MaxEnt model simulation (Fig. 3A) shows that the total suitable area for *B. minax* in Sichuan Basin is $21.61 \times 10^4$ km$^2$, accounting for 87.62% of the basin. Specifically, the highly-suitable area is $6.61 \times 10^4$ km$^2$, including most of the eastern parallel ridge valley, the north of the middle shallow hilly zone, the north and south of the peripheral hilly zone and the northeast of the Chengdu Plain. The moderately-suitable area is $9.34 \times 10^4$ km$^2$, including the middle and east of the middle shallow hilly zone the south of the eastern parallel ridge valley, the north of the peripheral hilly zone, and the middle and west of the Chengdu Plain. The rest is lowly-suitable area, which is mainly distributed in the west and north of the peripheral hilly zone, the southwest of the middle shallow hilly zone, the north of the eastern parallel ridge valley and the south of the Chengdu Plain. As can be seen from Fig. 3E, the total suitable area for *B. dorsalis* in Sichuan Basin is $23.49 \times 10^4$ km$^2$, accounting for 95.26% of the basin. Specifically, the highly-suitable area is $2.62 \times 10^4$ km$^2$, including most of the eastern parallel ridge valley, the southeast of the middle shallow hilly zone and the eastern of the peripheral hilly zone. The moderately-suitable area is $10.95 \times 10^4$ km$^2$, including the middle and east of the middle shallow hilly zone, the south of the eastern parallel ridge valley, the north of the peripheral hilly zone, and the middle and west of the Chengdu Plain. The rest is lowly-suitable area, which is mainly distributed in the west and north of the peripheral hilly zone, the southwest of the middle shallow hilly zone the north of the eastern parallel ridge valley and the south of Chengdu Plain.

The forecast distribution map (Figs. 3B, 3F) using the Garp model shows that besides the west of the peripheral hilly zone, the north and south of the Chengdu Plain, the remaining area is highly-suitable area for *B. minax*, accounting for 73.49% of the Sichuan Basin. The moderately-suitable area is $2.28 \times 10^4$ km$^2$, which can be divided into two parts, one part

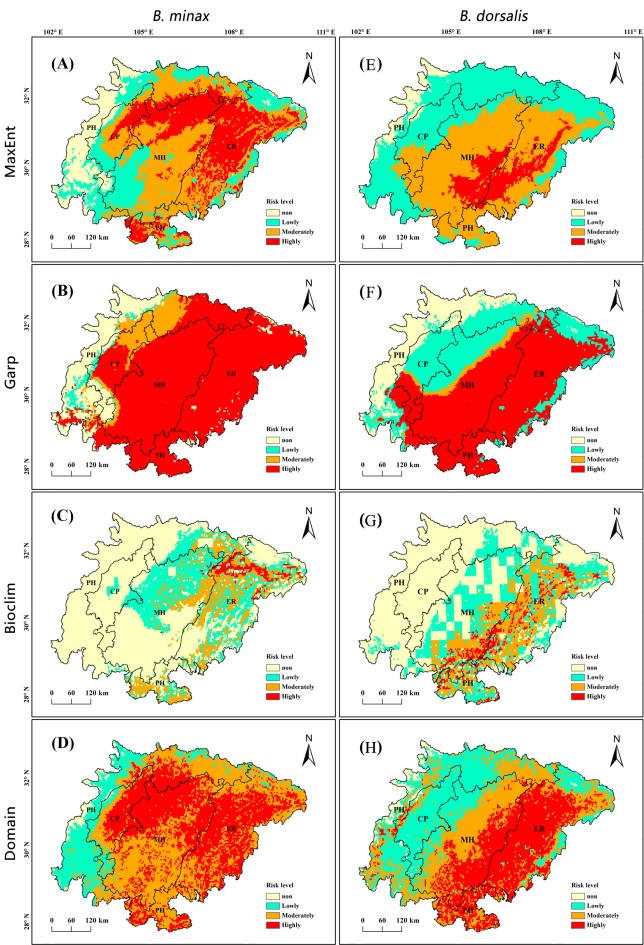

**Figure 3** **Potential distribution of *B. minax* (A–D) and *B. dorsalis* (E–H) simulated by MaxEnt, GARP, Bioclim and Domain.** CP, Chengdu plain; MH, middle shallow hilly zone in Sichuan Basin; ER, eastern parallel ridge valley in Sichuan Basin; PH, peripheral hilly zone in Sichuan Basin.

is concentrated in the north of the peripheral hilly zone and the north of the Chengdu Plain, and the other part extends from the south of the Chengdu Plain to the southwest of the middle shallow hilly zone in a strip shape. The lowly-suitable area is $0.57 \times 10^4$ km$^2$, which is scattered in the west of the peripheral hilly zone and the west of the Chengdu Plain. The highly-suitable area for *B. dorsalis* mainly concentrate in other areas except the northwest of middle shallow hilly zone, the north and southeast of eastern parallel ridge valley, the north of Chengdu plain and the southeast of peripheral hilly zone, and the total area is $18.12 \times 10^4$ km$^2$. The moderately-suitable area is mainly in the southwest margin of the high suitable area, with a gross area of $2.28 \times 10^4$ km$^2$. The unsuitable-area for *B. dorsalis* located in the peripheral hilly zone is $3.68 \times 10^4$ km$^2$, accounting for 15% of the basin.

According to the Bioclim (Figs. 3C, 3G) and Domain (Fig. 3D, 3H) obtained by the DIVA-GIS software, it can be observed that the red region representing the highly-suitable area accounts for a small proportion in these two models. Geographically, Bioclim

predicted a small distribution range, the highly-suitable area and moderately-suitable area for *B. minax* are located in the east of Sichuan Basin. The highly-suitable area ($0.59 \times 10^4$ km$^2$) is distributed in the north of the eastern parallel ridge valley in thin strips. The moderately-suitable area ($3.66 \times 10^4$ km$^2$) is not concentrated, one part is locate in the eastern parallel ridge valley and adjacent middle shallow hilly zone in a south-north strip shape, and the other is distributed in the north and south of the peripheral hilly zone in a block shape. While the highly-suitable area and moderately-suitable area for *B. dorsalis* are scattered located in east of middle shallow hilly zone, west of eastern parallel ridge valley and west of peripheral hilly zone, with a gross area of $5.66 \times 10^4$ km$^2$. The distribution range predicted by the Domain model is large, which is generally close to that of Garp model and the distribution range of highly and moderately suitable area are similar to that of Maxent.

## Relationship between probability of species and key environmental factors

### Screening key environmental factors

The importance of factors used in the modeling was analyzed again by the tool of jackknife test, and the analysis results are shown in Fig. 4. Considering the percent contribution and the permutation importance of environmental factors, it can be seen from the Fig. 4A that mean temperature of driest quarter (Bio9), annual precipitation (Bio12), temperature annual range (Bio7), precipitation of direst month (Bio14) and mean temperature of warmest quarter (Bio10) have high predictive ability, which the percent contribution and permutation importance are 83.96% and 87.87%, respectively. Therefore, these five factors are identified as the key environmental factors affecting the distribution of *B. minax*. Similarly, it can be seen from Fig. 4B that the key environmental factors affecting the distribution of *B. dorsalis* are mean temperature of driest quarter (Bio9), temperature seasonality (bio4), mean diurnal range (bio2) and precipitation of direst month (Bio14).

### The suitable range of key environmental factors

Figure 5 is the response curve between key environmental factors and distribution probability drawn by the Maxent model, which can reflect the value range of environmental factors under different thresholds. According to the *Wang et al. (2020b)* classification method, the study took 0.33 as the threshold to divide the range of environmental factors suitable for *B. minax* and *B. dorsalis*. The response curves of five key environmental factors affecting the potential distribution of *B. minax* (Figs. 5A–5E) demonstrate that the suitable ranges of temperature annual range, mean temperature of driest quarter, mean temperature of warmest quarter, annual precipitation and precipitation of driest month are 26.27–31.17 °C, 4.58–7.07 °C, 24.58–27.57 °C, 1,025.64–1,380.34 mm and 15.2–44.65 mm, respectively. Similarly, the key environmental factors affecting the potential distribution of *B. dorsalis* (Fig. 5F–5I) are mean diurnal range, mean temperature of driest quarter, temperature seasonality (standard deviation ×100) and precipitation of driest month, which the threshold ranges are 2– 8.6 °C, ≥13.1 °C, 341.88–666.67, 15.4–63.9 mm, respectively.
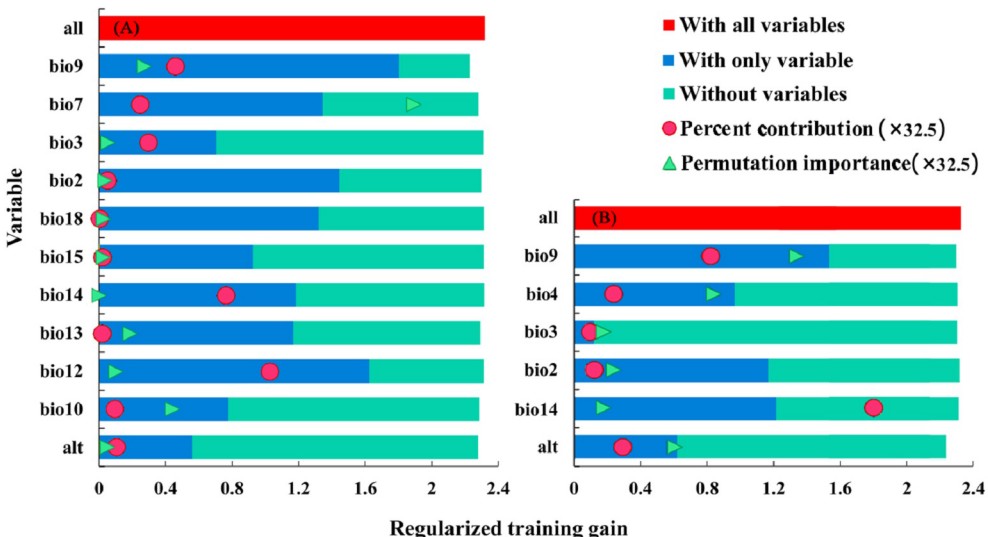

**Figure 4** The importance of environmental factors affecting the distribution of *B. minax* (A) and *B. dorsalis* (B) (jackknife).

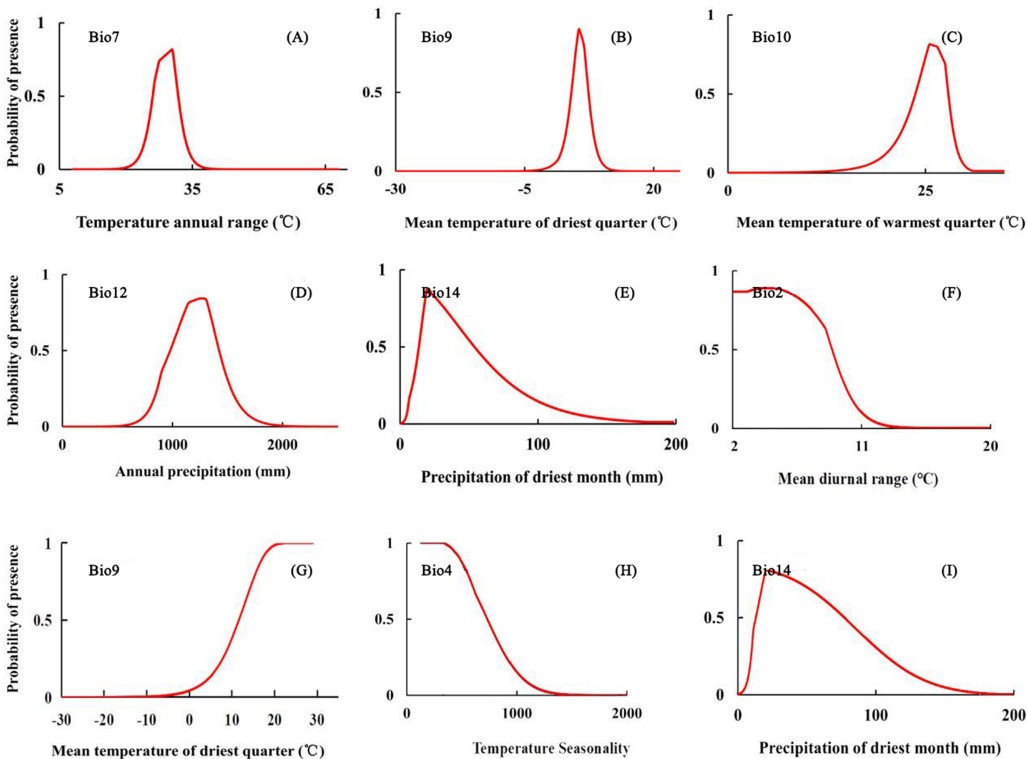

**Figure 5** Response curve of *B. minax* (A–E) and *B. dorsalis* (F–I) to key environmental factors.

## DISCUSSION

By comparison, it is found that the Maxent model can fit the distribution of *B. minax* and *B. dorsalis* in Sichuan Basin well, which the average value of AUC and Kappa are higher than other models. Relevant literature also proves that Maxent could predict the suitable area of species well under both large or small sample conditions (*Xu, Peng & Peng, 2015*). The Garp model predicts a wide range of highly-suitable area for *B. minax* and *B. dorsalis*, which is similar to what other scholars have done in predicting other species (*Wang et al., 2020a*; *Zhang et al., 2023*). This may be due to two reasons: one is that the output type of Garp is boolean, and the result is suitable as long as species shows adaptation somewhere, and another may be due to insufficient repetitive settings; the classification is not obvious. The Domain model uses the Gower algorithm to calculate the similarity between points, and uses similarity matrix to classify the study region (*Carpenter, Gillison & Winter, 1993*). The prediction results of the Domain model are greatly affected by the distribution points, and almost all the places with distribution points have certain suitable regions (*Zhang et al., 2017*). Based on the above reasons, the range of suitable area predicted by the Domain model is also large. The Bioclim defines the environmental envelope as the volume of a straight line in Euclidean space and treats each climate axis independently depending on the range, which can lead to ecologically unreasonable predictions (*Zhang et al., 2023*). Therefore, the Bioclim model is poor in predicting the distribution of *B. minax* and *B. dorsalis*.

The relationship between species and environment is an important aspect to study the ecological needs and spatial distribution of species. The article analyzes the relationship between the presence probability of *B. minax* and *B. dorsalis*, and the key environmental factors affecting the potential distribution of these two pests, and obtains the feedback curve. Among the five key factors affecting the distribution of *B. minax*, mean temperature of driest quarter and annual precipitation are the most important environmental factors, indicating that the diffusion and reproduction process of *B. minax* is restricted by the temperature and precipitation. In this study, the suitable range of mean temperature of driest quarter for *B. minax* is 4.58–7.07 °C, and the optimal value is 5.55 °C. The Sichuan Basin belongs to the subtropical humid climate zone, and the driest season is from November to April of the next year, *B. minax* overwinters as the pupae during this period. *Ma (2014)* found that when the temperature was between 5 °C and 10 °C, the pupae did not emerge and the survival time could reach 156–250 days, which was basically consistent with the temperature range in this study. The annual precipitation is related to the air and soil moisture, which has a significant effect on the emergence and survival of imago. When the soil water content is low or high, the emergence rate is significantly inhibited and the mortality rate increases. Among the four key environmental factors affecting the distribution of *B. dorsalis,* mean temperature of driest quarter and precipitation of driest month are the most important environmental factors. Studies have shown that *B. dorsalis* is a tropical and subtropical insect, and its occurrence and distribution are greatly affected by climate conditions such as temperature and humidity. The developmental threshold temperature of pupae is 9∼11 °C, and neither too low nor too high temperature can

make pupae survive or emerge safely (*Kong et al., 2008*). The influence of precipitation on *B. dorsalis* is mainly manifested in two aspects under the natural conditions. On the one hand, appropriate precipitation can maintain soil moisture and atmospheric moisture, thus reducing the mortality of mature larvae and newly emerging adults, which is conducive to the mating and spawning activities of adults. On the other hand, excessive precipitation will cause high soil moisture, which will affect the larva pupae and pupae grow up to imago. It should be noted that the feedback curve between presence probability of pests and environment factors reflects the effect of the single environment factor, but the life activities of pests is affected by a variety of environmental factors (including climate factors, host condition, natural enemy species, vegetation, *etc.*). Therefore, this result can be used as a reference to judge the relationship between pests and environmental factors, but it can not completely explain the relationship between them.

The risk of detecting exotic pests late or unresponsive can be illustrated by cases of eradication failure, such as *B. carambolae* in Suriname, the lag phase from the first discovery of infested fruit in 1975 to the identificaton and confirmation of specimens from South-east Asia in 1986 was 12-year (*Dias et al., 2018*; *van Sauers-Muller, 2005*). SDMs can predict the possible range of pests before actual problems occur, and make pre-emptive and effective pest management decisions. The research shows that the suitable area for *B. minax* is mainly concentrated in the eastern of the Sichuan Basin, while the *B. dorsalis'* is concentrated in the southeastern. Except for the Bioclim model, the highly-suitable area and moderately-suitable area predicted by the other three models are larger than $15.94 \times 10^4$ km$^2$ (accounting for 55.1% of the Sichuan Basin) and $13.58 \times 10^4$ km$^2$ (accounting for 65% of the Sichuan Basin), indicating *B. minax* and *B. dorsalis* are suitable for survival in the Sichuan Basin.

Prevention is one of the most effective strategies for controlling and monitoring of pests, and is crucial for determining the population dynamics, comparing infestation levels among different species, and evaluating the effectiveness of control strategies (*Eliopoulos, 2007*). With the continuous development and improvement of analytical biotechnology and biochemical technology, many new techniques have been applied in the identification of insect classification. Brazilian researchers had proposed a multimodal fusion method based on two types of images (wings and aculei), which pointed the way to identify *Anastrepha*. Now an algorithm had been developed to identify blotches in hyperspectral images of mangoes that had been invaded by tephritidae larvae. Meanwhile, the development of automatic insect traps has been strengthened and accelerated. According to relevant studies, the above methods can be used to monitor pests well. Therefor, we should strengthen the study of *B. minax* and *B. dorsalis* in the Sichuan Basin on the basis of the above research.

The spatial distribution of species reflects the inter-relationship between population individuals in horizontal space, and is the result of long-term adaptation and selection between population and environment. The study of population spatial pattern can reflect a series of ecological processes that plant populations have experienced in the past. Therefore, the prediction of the potential geographical distribution of *B. minax* and *B. dorsalis* by four models can understand the population characteristics, the intra-specific and inter-specific relationships, and the relationship between the population and the environment, so as

to study and develop feasible quarantine pest control countermeasures. It is of great significance to reduce the serious economic losses caused by quarantine pests and ensure the stable development of China's fruit, vegetable and flower industry. The combined results of the four models find that the northern part of middle shallow hilly zone and the eastern part of parallel ridge valley in Sichuan Basin are high risk area for the occurrence of *B. minax*, while the eastern part of middle shallow hilly zone and the western part of parallel ridge valley in Sichuan Basin are high risk area for the occurrence of *B. dorsalis*, so these areas should take preventive measures to prevent *B. minax* and *B. dorsalis* further spread and outbreak. The prediction results show that there are scattered distribution records of *B. minax* and *B. dorsalis* in peripheral hilly zone of the Sichuan Basin, so it is recommended to carry out census work to find out the actual distribution and establish early monitoring and warning system. At the same time, the establishment of a rapid response and real-time control system for the epidemic and the strengthening of popular science publicity are also conducive to "early detection, early warning and early response".

As the main group of animals, the distribution of insects is greatly affected by the environment and host distribution (*Xin et al., 2019*). Due to data limitations, the study only considered meteorology and altitude affecting the suitable areas of *B. minax* and *B. dorsalis*, which might cause the predicted ecological niche to be larger than the actual ecological niche. In the next step, we will focus on considering the reliable expression of the interaction of various factors to improve the prediction effect of the model.

### Funding
This work was funded by the Natural Science Foundation of Sichuan Province (2022NSFSC0589), the Sichuan Genuine Medicinal Materials and Traditional Chinese Medicine Innovation Team (SCCXTD-2022-19), the Sichuan Science and Technology Program (2021YFYZ0012), the Key R & D projects of the Science and Technology Deportment of Sichuan Province (2022YFS0592) and the Heavy Rain and Drought-Flood Disasters in Plateau and Basin Key Laboratory of Sichuan Province (SCQXKJYJXZD202209). The funders had no role in study design, data collection and analysis, decision to publish, or preparation of the manuscript.

### Grant Disclosures
The following grant information was disclosed by the authors:
Natural Science Foundation of Sichuan Province: 2022NSFSC0589.
Sichuan Genuine Medicinal Materials and Traditional Chinese Medicine Innovation Team: SCCXTD-2022-19.
Sichuan Science and Technology Program: 2021YFYZ0012.
Science and Technology Deportment of Sichuan Province: 2022YFS0592.
Heavy Rain and Drought-Flood Disasters in Plateau.
Basin Key Laboratory of Sichuan Province: SCQXKJYJXZD202209.

## Competing Interests

The authors declare there are no competing interests.

## Author Contributions

- Yanli Xia performed the experiments, prepared figures and/or tables, and approved the final draft.
- Jinpeng Zhao conceived and designed the experiments, prepared figures and/or tables, authored or reviewed drafts of the article, and approved the final draft.
- Jian Ding conceived and designed the experiments, authored or reviewed drafts of the article, and approved the final draft.
- Ke Xu conceived and designed the experiments, authored or reviewed drafts of the article, and approved the final draft.
- Xianjian Zhou analyzed the data, prepared figures and/or tables, and approved the final draft.
- Mian Xiang analyzed the data, prepared figures and/or tables, and approved the final draft.
- Huiling Xue analyzed the data, prepared figures and/or tables, and approved the final draft.
- Huan Wang analyzed the data, prepared figures and/or tables, and approved the final draft.
- Rulin Wang analyzed the data, prepared figures and/or tables, and approved the final draft.
- Yuxia Yang analyzed the data, authored or reviewed drafts of the article, and approved the final draft.

## Data Availability

The raw measurements are available in the Supplementary Files.

## Supplemental Information

Supplemental information for this article can be found online at http://dx.doi.org/10.7717/peerj.16745#supplemental-information.

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
