# Peer review of "Geographical distribution of two major quarantine fruit flies (Bactrocera minax Enderlein and Bactrocera dorsalis Hendel) in Sichuan Basin based on four SDMs"

_PeerJ, doi:10.7717/peerj.16745_

## Round 0.1 · original submission · Major Revisions

Kindly address all the comments, especially the "Major Concerns" raised regarding number of data points used in the study, data processing, implications, and potential limitations of the study.

Reviewer 1 ·

Basic reporting

A comprehensive understanding of theirs current spatial patterns is crucial for co-management. The work presented in this manuscript is a straight forward study. However, the paper suffers from several shortcomings as detailed in the follows and needs corrections.

Experimental design

1. The species distribution data are all taken from an online database, and the time of access to the data must be specified. The distribution data must be filtered by spatial correlation analysis, which can be done with the R package "spThin". Only climatic variables have been chosen for the environmental variables. The topography, soils, hydrology and human activities should also be considered. VIF is used more reasonably in dealing with multicollinearity than the correlation coefficient.
2. The simulation method is not standard, there is no new contribution in terms of model or method development. For instance, there are many other parameters in Maxent model and why did the authors only pick those parameters? Please note that AUC is a very criticized evaluation metric in ENMs studies. There were two occurrence-based metrics launched in 2018 for SDMs that could be used instead of AUC. See Leroy et al. 2018. (Leroy B, Delsol R, Hugueny B, et al (2018) Without quality presence-absence data, discrimination metrics such as TSS can be misleading measures of model performance. J Biogeogr 45:1994-2002. doi: 10.1111/jbi.13402). “Feature Classes” (FCs) and “Regularization Multiplier” (RM) are the most significant parameters of the Maxent model and can affect the accuracy of the model outputs. Their calibrations can significantly improve the accuracy of the Maxent model outputs. In the present study, I suggest you make the calibration of Maxent with ENMeval package in R. Muscarella, R., Galante, P.J., Soley-Guardia, M., Boria, R.A., Kass, J.M., Uriarte, M. and Anderson, R.P. (2014), ENMeval: An R package for conducting spatially independent evaluations and estimating optimal model complexity for Maxent ecological niche models. Methods Ecol Evol, 5: 1198-1205. https://doi.org/10.1111/2041-210X.12261.
3. Introduction: It should follow the state of the art in this field and review what has been done, for supporting the research gap and the significance of this review. Please, improve the state-of-the-art overview, to clearly show the progress beyond the state of the art. The lack of proper justification creates the wrong impression that the authors are unaware of the recent developments. Please reason both the novelty and the relevance of your review goals.
4. The method description is not clear. For example, what are the environment variables selected? What is the basis for the selection? The basis, method, and validation of the species distribution model should be described in detail as necessary.
5. The results should be analyzed and discussed in more detail and depth. The current manuscript does not reflect the connection between the results of the study and the actual distribution of the species very well.
6. The practical guidance implications of the study for species management should be analyzed in relation to the results of the study.

Validity of the findings

no comment

Additional comments

no comment

Reviewer 2 ·

Basic reporting

1) the English is fine enough though a few areas need improvement (see pdf for suggestions in several areas)
2) The references throughout the manuscript should be improved. First, grey literature (non peer-review) should be avoided. For instance, I looked up Zhu 2004 using the title but found Asokan et al. 2012 with the exact same title.
Second, several of the references are about herbal medicines. I have no idea how that relates to this paper.

For examples…
Deng 2018. Research progress of molecular mechanism of pharmacological effects of Fuzzi
Li, Y., Li, W., Wang, K.L., Liu, Q.C., Jiang, X.Q., Liu, Q.H., 2017. Cloning and expression analysis of DFR gene in 343 Aconitum carmichaeli. Plant Physiology Communications 53, 2222-2228.
Li, S., Li, R., Zeng, Y., Meng, X.L., B, W.C., Zheng, S.C., 2019. Chemical components and pharmacological action 345 of Aconiti Radix. China journal of Chinese Maeria Medica
and many others…

3) The structure is fine. Raw data are shared in supplement. Figures and tables fine thought table 2 could be deleted.

4) the study is self-contained

Experimental design

1) the article is within the Aims/scope
2) This manuscript focuses on modelling the species distribution of two tephritids in China. The overall objective of the manuscript is clear in that it seeks to provide insight on potential species ranges in China.
3) The approach undertaken in this study is fairly standard (MAXENT, Bioclim, GARP, Domain). The results are a set of distribution maps
4) There are a few things the methods need. First, the sample sizes need to be presented as well as a map showing the occurrence data so that the reader has an idea of spatial range. Second, there needs to be a data analysis section explain the modelling steps, software used, and specifics to each model.

Validity of the findings

1) All underlying data has been provided.
2) distributions maps are produced.

Additional comments

There are a few major weaknesses with the manuscript in its current form.

Introduction – This section needed to improved. First, more background information on the two main species need to be provided. I had no clue if they are endemic to China or recent invaders. If endemic, then does SDM make sense as the current distribution indicates it presence. If recent invader, how well does the current distribution reflect its true niche range. Might it better to use other locations and then extrapolate to China? Second, the introduction needs a final paragraph that better introduces what this study will be doing. I just got lost in the final paragraph on what paper was what and if the authors were referring to this study or other.

Discussion – This section is okay. It could be improved by discussing potential limitations of the models, comparing and contrasting the strengths and explain why Bioclim model predicted very different distributions. Also, I think that the authors mixed up Drosophila and Tephritids in the final paragraph [maybe they just wanted to rile up entomologist].

I have included minor comments in the attached pdf.

Annotated reviews are not available for download in order to protect the identity of reviewers who chose to remain anonymous.

·

Basic reporting

The manuscript is fairly well written. There is a need for additional information on the biology of the two pests. The hypothesis, scientific problem, and objectives of the study are not clearly stated. So the study is not hypothesis-driven. Moreover, there are formatting issues that need to be addressed in the revised version.

Experimental design

The work is original, but the research question is not properly formulated. There is a need for additional information on the software used for the calculation of the suitable areas and how the calculation was carried out is missing.

Validity of the findings

The performance model was assessed using AUC and Kappa. However, the methodology section requires additional information.

Additional comments

The authors investigated the geographical distribution of two major quarantine fruit flies (Bactrocera minax Enderlein and Bactrocera dorsalis Hendel) in the Sichuan Basin based on four SDMs. The work is important, fairly well-written, and provides interesting results. However, there are concerns that need to be addressed to strengthen the MS;

Minor concerns

1. The scientific problem and hypothesis for the study should be clearly stated in the background section
2. AUC and Kappa were used to assess the performance of the models
3. Please check the spacing in line 35
4. Line 35: please it’s important to mention the exact figure here rather than being too generic. …..104. Note that an abstract should stand alone and shouldn’t create any doubt
5. Line 39: It is important the “keywords’ help readers capture the idea of the work. Preferably;
Species distribution models, Bioclim, Domain, GARP, MaxEnt, suitable habitat, B. dorsalis, B. minax
6. Lines 42-87, the study is not hypothesis-driven and the scientific problem making the work compelling is not clearly stated. The last paragraph of the introduction should clearly capture the scientific problem and hypothesis and objectives of the study
7. Check for consistency in the font size of the legends in Figure 4.
8. Check the number of figures. It seems you interchanged the Figure 1 and 2 legends. Please correct it
9. There formatting issues that need to be corrected throughout the manuscript. Please refer to the pdf file for some of them.
10. The authors need to mention what was used for the calculation of the suitable areas and how it was done

Main concern
11. How many occurrence records were obtained from the survey before the cleaning process and how many were left after the cleaning process. This should be stated in the methodology section.
12. In the introduction section, there should be a paragraph on the biology of the two fruit flies
13. What is the implication of your study for the management of the two species? It should be clearly captured as a paragraph in discussion section
14. This should follow by another paragraph explaining the limitations of the study.

---

## Round 0.2 · Minor Revisions

Kindly address the issues raised by the reviewer specifically the comments on the validity of the paper findings. Also, please have the paper checked for grammatical errors.

**Language Note:** The Academic Editor has identified that the English language must be improved. PeerJ can provide language editing services - please contact us at [email protected] for pricing (be sure to provide your manuscript number and title). Alternatively, you should make your own arrangements to improve the language quality and provide details in your response letter. – PeerJ Staff

Reviewer 2 ·

Basic reporting

1) Again, there are some grammar issues in the text but nothing overly problematic. I noted a few that jumped out in the annotated pdf.

2) The literature looks better.

3) The article meets basic professional standards with structure.

4) The authors provide some basis for why the environmental factors are chosen in relation to the species ecology

Experimental design

The authors has improved the manuscript as suggested and that has helped with following the methods and making it more reproducible.

The purpose of the study is clear.

The results get a bit dense with what I consider un-useful estimates of areas for lots of regions. I would have preferred more focus on difference in total areas of the different suitability classes and compare/contrast the different predictors driving the relationships.

Validity of the findings

The authors need to better justify the strong focus on MaxEnt results. The authors report the AUC incorrectly either in the results or in the discussion (if the latter it affects their logic for preferring the MaxEnt...I assume they prefer given the focus on those relationships)

The main conclusion is that most models suggest a potentially large region and provide a starting point for decision making.

Annotated reviews are not available for download in order to protect the identity of reviewers who chose to remain anonymous.

---

## Round 0.3 · Minor Revisions

The authors have satisfactorily addressed the reviewers' comments and is ready for publication from a scientific standpoint. However, there are still a number of language/grammatical errors present in the manuscript. Please have the manuscript undergo language editing.

**Language Note:** The Academic Editor has identified that the English language must be improved. PeerJ can provide language editing services - please contact us at [email protected] for pricing (be sure to provide your manuscript number and title). Alternatively, you should make your own arrangements to improve the language quality and provide details in your response letter. – PeerJ Staff

---

## Round 0.4 · Minor Revisions

Most of the grammatical revisions were on verbs (i.e., tenses). Please also considered improving the sentences structures to clearly convey the message in some of the statements. I suggest that the authors have the manuscript edited thoroughly.

**Language Note:** The Academic Editor has identified that the English language must be improved. PeerJ can provide language editing services - please contact us at [email protected] for pricing (be sure to provide your manuscript number and title). Alternatively, you should make your own arrangements to improve the language quality and provide details in your response letter. – PeerJ Staff

---

## Round 0.5 · accepted · Accept

Thank you for revising the manuscript and congratulations